

**Mixing state of refractory black carbon at different atmospheres in China**
Gang Zhao[1], Tianyi Tan[1], Shuya Hu[1], Zhuofei Du[1], Dongjie Shang[1], Zhijun Wu[1,2],
Song Guo[1,2], Jing Zheng[1], Wenfei Zhu[1], Mengren Li[1], Limin Zeng[1], Min Hu[1,2*]
1 State Key Joint Laboratory of Environmental Simulation and Pollution Control,
International Joint Laboratory for Regional Pollution Control, Ministry of Education,
College of Environmental Sciences and Engineering, Peking University, Beijing,
100871, China
2 Collaborative Innovation Center of Atmospheric Environment and Equipment
Technology, Nanjing University of Information Science & Technology, Nanjing,
China
*Correspondence author:** Min Hu (minhu@pku.edu.cn)
**Abstract**
Black carbon (BC) particles exert a significant influence on the earth's climate
system. However, large uncertainties remain when estimating the radiative forcing by
BC because the corresponding microphysical properties have not been well addressed.
Knowleadge of the BC mixing states of different aging degree can help better
characterise the corresponding environmental and climate effects. In this study, the BC
size distributions were studied based on three different field campaigns at an urban site,
a suburban site, and a background site in China using a single particle soot photometer
(SP2) in tandem with a differential mobility diameter. Measurements from the SP2
indicates that the BC particles were composed of either fresh or aged aerosols. The
mean number fractions of the fresh BC aerosols were 51%, 67%, and 21% for the





urban, suburban, and background sites, respectively. The corresponding mobility
diameters of these aged (fresh) BC-containing aerosols were 294 nm (193 nm), 244
nm (161 (nm), and 257 nm (162 nm). The measured aged (fresh) BC core number
median diameters were 115 nm (114 nm), 107 nm (95 nm), and 127 nm (111 nm) for
urban, suburban, and background sites, respectively. The corresponding aged (fresh)
core mass median diameters were 187 nm (154 nm), 182 nm (146 nm), and 238 nm
(163 nm) respectively. The mean diameter of the aged BC-containing aerosols was
larger than that of the fresh BC-containing aerosols, while the mean BC core diameter
of the aged BC-containing aerosols was smaller than that of the fresh BC-containing
aerosols. About 10% of the BC-containing aerosols with the BC core were attached to
the other non-BC components, which were mainly generated by coagulation between
the BC and non-BC components. The measurement results in our study can help better
understand the BC size distributions and mixing status in the different atmospheres in
China and can be further used in modeling studies to help constrain the uncertainties
of the BC radiative effects.

**Introduction**


Black carbon (BC) plays an important role in the climate system by absorbing
solar radiation (Ramanathan et al., 2008), interacting with the cloud (Roberts et al.,
2008), and changing the albedo of the snow (Menon et al., 2002). It is the second most
important aerosol component after carbon dioxide, contributing to global warming
(Bond et al., 2013). The solar absorption of BC has a significant influence on the
development of the boundary layer and then aggravates the air pollution (Ding et al.,
2016). The turbulence in the atmospheric boundary layer can be suppressed due to the
existence of BC (Wilcox et al., 2016). The BC also plays a remarkable role in driving
the formation and trend of regional haze (Zhang et al., 2020).



BC is mainly generated by the incomplete combustion of biofuels and fossil fuels (Bond et al., 2006). After emission, the morphology of BC transforms from fractal to spherical and subsequently grows to a fully compact particle with other chemical components coating on it (Peng et al., 2016). During the aging process, the BC optical properties change significantly up to a factor of 3 and then the corresponding magnitude of climate forcing contributed by BC is increased by up to a factor of 2 (Zhang et al., 2008; Cappa et al., 2012). Large uncertainties remain in estimating the BC radiative effects due to the large variation in BC microphysical properties, such as size distributions and mixing states during the aging process (Moffet et al., 2016; Matsui et al., 2018; Zhao et al., 2019). Therefore, characterizing the differences in size distributions and mixing states between the fresh and aged BC particles can help better constrain the uncertainties of BC aerosol radiative effects. To our best understanding, few studies have specified the mixing states and size distributions of both the fresh and aged BC aerosols.

The BC-containing particles can also be classified into two morphological types: bare BC on the surface of non-BC particles (attached type) and BC embedded within or coated by non-BC components (coated type). With the same amount of non-BC components, the mass absorption cross-sections of BC by the attached type are much smaller than those by the coated type (Moteki et al., 2008; Moteki et al., 2010a; Moteki et al., 2014). Therefore, the impact of BC on climate can be better estimated when accurately identifying the two types of ambient BC-containing particles. Observations are required to constrain the spatial and temporal microphysical properties of the atmospheric BC.

The single-particle soot photometer (SP2) is always used to measure the mixing states and size distributions of ambient BC particles. The measured signals from SP2 can be used to distinguish the BC-containing aerosols as fresh thinly and aged thickly





coated ones. The measured results can also be employed to distinguish the BC-containing particles between attached and coated types, which were described in detail in the methodology part.

In this study, the tandem SP2 and differential mobility analyzer (DMA) was employed at an urban site, a suburban site, and a background site in China to investigate the microphysical properties of the BC particles. The size distributions and mixing states of both the fresh and aged BC aerosols at different atmospheres were characterized. We also investigated the corresponding morphology properties of the BC-containing aerosols. The measured microphysical properties provide the basis for future modeling studies of the BC radiative effects of different environment in China.

## 2 Methodology

### 2.1 Measurement sites

The measurements were conducted at three different atmospheric sites in China, namely the urban site of Peking University Urban Atmosphere Environment Monitoring Station (PKU, 39.9°N,116.1°E, 58m a.s.l) in Beijing between 20 January and 4 February 2016, the suburban site of Changping (CP, 40.3°N,116.2°E, 70m a.s.l)) in Beijing between 15 May and 5 June 2016, and the background site of Lijiang (LJ, 27.2°N,100.2°E, 3410 m a.s.l) in Yunnan Province between 22 March and 4 April 2015. The PKU site is located in the northwest of Beijing. This site could characterize the air pollution of the urban Beijing (Hu et al., 2017; Hu et al., 2021). The CP site locates at the northwest of the Beijing urban area, representing a regional atmosphere (Wang et al., 2019b; Zhao et al., 2021). The LJ site represents the background areas, located in the Mountain Yulong, in the Yunan Province of China (Zheng et al., 2017; Shang et al., 2018; Wang et al., 2019a). The aerosol optical depth at the wavelength of 550 nm during the year 2020 indicated that the LJ site was very





clean and the PKU and CP sites were more polluted as shown in Fig. S1 in the
supplement.

## 2.2 Instruments

### 2.2.1 DMA-SP2 system

As for the SP2, the continuous Nd: YAG laser beam with the wavelength of 1064
nm is generated intensively in the instrument chamber. When the BC-containing
particles pass through the laser beam, they absorb the radiation and then are heated to
around 3500-5000 K. The intensity of the emitted incandescent light from the heated
BC particle is then transformed to the BC mass concentration. The scattering signals
of the BC particle are recorded to estimate the BC particle mixing state.
In this study, the SP2 (Droplet Measurement Technology, Inc., USA) was placed
after the DMA (Model 3081, TSI, USA) to measure the size-resolved BC mixing
states, and the instrument setup is schematically shown in Fig. S2. The DMA was set
to scan the aerosol over the size range between 12.3 and 697 nm every five minutes.
The flow rate leading to the SP2 and the condensation particle counter (CPC, Model
3776, TSI, USA) were 0.12 and 0.28 L/min, respectively. The sheath flow of the DMA
was 4 L/min.
The Aquadag was used to calibrate the measured incandescence signal of the SP2
using the DMA-SP2 system. The formula from Gysel et al. (2011) was used to convert
the mobility diameter into the mass of Aquadag. A correction factor of 0.75 was
applied to account for the different response sensitivity of SP2 to Aquadag and
ambient BC (Moteki et al., 2010b).


In this study, the coating thickness of the BC-containing aerosols was calculated
by the difference between the total mobility diameter measured by the DMA and the
mass equivalent diameters of the BC core with the assumption that the density of the
BC-core is 1.8 g/cm$^3$.
**2.2.2 Other instruments**
The submicron particles (PM$_1$) chemical compositions were measured using a
high-resolution time-of-flight aerosol mass spectrometer (AMS; Aerodyne Research
Inc., Billerica, MA, USA). The data processing software PIKA (version 1.16) was
used for data analysis. The positive matrix factorization (PMF) analysis was conducted
for source appointment of the organic aerosols (Ulbrich et al., 2009). More details of
the measurement of the aerosol chemical compositions and data processing can be
found in Zheng et al. (2017).
The mass concentrations of O$_3$ were measured using UV absorption (model 49i,
Thermo Fischer Inc. USA) with a time resolution of 1 minute. The mass
concentrations of NO and NO$_2$ were measured using the chemiluminescence technique
(NO-NO$_2$-NO$_x$ Analyzer, Model 42i, Thermo Scientific, USA). The mass
concentrations of SO$_2$ were measured using the ultraviolet fluorescence method (SO$_2$
analyzer, model 43i-TLE, Thermo Scientific, USA). The temperature (T), relative
humidity (RH), wind speed (WS), and wind direction (WD) were monitored
continuously during these campaigns.
**2.3 Methodology**
For the BC-containing aerosol, there is a lag between the peak time of the
scattering and the incandescence signal (Metcalf et al., 2012). The lag time between
the peak scattering signal and the peak incandescence signal can be employed to


describe the coating thickness (Schwarz et al., 2006; Moteki et al., 2007) and further
used to distinguish the BC-containing aerosols as fresh thinly and aged thickly coated
ones.
The measured scattering and incandescence signal can also be employed to
distinguish the BC-containing particles as attached and coated types (Moteki et al.,
2014) by calculating the time-dependent scattering cross-sections of BC-containing
particles (Moteki et al., 2007). For the coated type, all of the coating material will
evaporate and the scattering cross-sections will decrease to zero after passing through
the laser beam, while the scattering cross-section of the attached BC-containing
aerosol will not decrease to zero (Moteki et al., 2008). The method adopted by
Dahlkötter et al. (2014) was employed here to characterize the morphology of the
BC-containing aerosols.
**3 Results and discussions**
**3.1 Overview of the measurement results at different atmospheres**
The time series of the measurement results are shown in Fig. S3, Fig. S4, and Fig.
S5 for the PKU, CP, and LJ sites, respectively.
As for the PKU site, the wind was mainly from the north and the wind speed was
low with a mean value of 2.2 m/s. The ambient atmosphere was very dry with a mean
RH of 27.6%, with minimum and maximum values of 5.8% and 72.6%, respectively.
The temperature in the winter of Beijing had a mean value of 0.8 $^{\circ}$C between -5.9 $^{\circ}$C
and 9.2 $^{\circ}$C. The mean mass concentration of $PM_{2.5}$ was 49.3 $\pm$ 55.4 $\mu g/m^3$. The
concentration of $SO_2$ and $NO_x$ ($NO_x$=NO + $NO_2$) had the same trends as $PM_{2.5}$, with
mean values of 16.3 $\pm$ 11.9 ppb and 68.2 $\pm$ 63.4 ppb, respectively. The $O_3$


concentration is anti-correlated with $PM_{2.5}$. The measurement site experienced four
main pollution periods between 20, January and 4, February, with each period lasting
2~4 days. The four pollution periods happened from 21 January to 24 January, from
24 January to 26 January, from 28 January to 29 January, and from 31 January to 3
February. The $PM_{2.5}$ peaked at the first pollution period, with 272.8 $\mu g/m^3$. For each
period, the high RH and low wind speed favored the development of pollution. At the
end of each pollution period, the $PM_{2.5}$ dropped dramatically with the increment of the
wind speed and the change of the wind direction. The environment in the winter of
Beijing was polluted, which was highly influenced by both primary particle emissions
and secondary formation influenced by the meteorology conditions.
For the suburban site CP, the wind showed obvious diurnal cycles with high-speed
west wind during the day and low-speed east wind during the night. The mean wind
speed was 2.4 ± 1.6 m/s. The RH during the campaign was 38.8 ± 16.0%, with a
maximum value of 80.5%. The temperature during the campaign was 21.8 ± 5.2 $^o$C
with a maximum value of 33.2 $^o$C. As for the $NO_x$, the mean concentration was 21.4 ±
17.7 ppb. The concentration of $NO_x$ experienced high value during the early morning,
and fluctuated dramatically, which is highly related to the anthropogenic activities.
The mean concentration of $SO_2$ was 2.89 ± 1.10 ppb. The measured $SO_2$ concentration
values during the day were higher than those at night. There was no obvious diurnal
cycle for the $SO_2$ concentration, and dramatic fluctuation was not observed, which
indicates that the $SO_2$ was mainly from transportation. The measured mean $O_3$
concentration was 54.5 ± 38.8 ppb. The mean $PM_{2.5}$ concentration was 22.6 ± 16.8
$\mu g/m^3$, with a maximum value of 71.8 $\mu g/m^3$.



As for the background LJ site, The mean value of the wind speed, RH, and T were
3.13 m/s, 50.23%, and 6.5 °C, respectively. The mean $PM_{2.5}$ mass concentration was
6.2 ± 5.7 μg/m³. The mean $NO_x$ and $SO_2$ concentrations were 0.05 ppb and 0.97 ppb
respectively.
The characteristics of the measurement sites are summarized and shown in Fig. 1.
The differences in the temperature and RH among these sites were mainly resulted
from the that the measurements were conducted in different seasons. The
concentrations of $SO_2$, $NO_x$, and $PM_{2.5}$ indicated that the urban site PKU was most
polluted. The suburban site CP was slightly polluted and the background LJ was the
cleanest.

**3.2 Mixing states of the fresh and aged BC-containing aerosols**

The measured lag time probability distributions for the PKU, CP and LJ sites are
shown in Fig. 2 (a), (b), and (c), respectively. The lag time had two modes for each
measurement site. In this study, the BC-containing aerosols with a lag time larger than
1.4 μs were classified as aged thickly coated particles. The other BC-containing
aerosols were classified as fresh thinly coated particles. Our critical lag time of 1.4 μs
is smaller than the previous studies that distinguished the BC-containing aerosols
between fresh BC and aged BC with a lag time of 2 μs (Moteki et al., 2007; Metcalf et
al., 2012) or 1.8 μs (Metcalf et al., 2012), which was determined by the internal setup
up of the SP2.
For each type of BC-containing aerosols, we calculated the coating thickness
probabilities and the results are shown in Fig. (d), (e) and (f) for the PKU, CP, and LJ
sites, respectively. Results showed that the BC-containing aerosols were mainly



composed of thickly coated aged BC aerosols and thinly coated fresh BC aerosols. The
coating thickness for the fresh BC-containing aerosol was smaller than that of the aged
BC-containing aerosols. However, the coating thickness of the aged BC-containing
aerosols spread wider than that of the fresh ones.
The number fractions of the aged BC-containing aerosols were significantly
different for different atmospheres as shown in Fig. 2 (g), (h), and (i). At the polluted
urban site, the number concentration of the aged BC-containing aerosols was
comparable to that of the fresh BC-containing aerosols with the number fractions of
51% and 49% for the fresh and aged BC particles, respectively. The number fraction
of the aged BC aerosols at the CP site was 67 %. However, the BC-containing aerosols
at the background LJ site were dominated by aged ones with a number fraction of

225   79%.

The difference in the number fraction of the aged BC particles was synthetically
influenced by the ambient pollution levels and the sources of the BC aerosols. The
suburban site CP had the largest number fraction of the fresh BC particles. The CP site
is not far from the urban, and thus the fresh BC particles from the traffic contribute a
large amount of the total ones. The urban site PKU had a larger number fraction of the
aged BC than that of the CP site. This might be resulted from that the PKU site being
more polluted than the CP site and then the aging processing at the PKU site was
faster than that at the CP site. The LJ site is far from the traffic sources. The measured
BC particles at the LJ site were mainly from long-range transportation and
experienced a long time of aging process than that at the CP and PKU sites. Therefore,
the BC-containing aerosols were dominated by the aged ones at the LJ sites.
We compared the number fraction of the aged BC at different measurement sites
from literature (Shiraiwa et al., 2007; Schwarz et al., 2008a; Schwarz et al., 2008b;





Subramanian et al., 2010; Huang et al., 2012; McMeeking et al., 2012; Metcalf et al.,
2012; Holder et al., 2014; Wang et al., 2014; Ueda et al., 2016; Wang et al., 2016;
Wang et al., 2017a; Wang et al., 2017b; Wang et al., 2017c; Wu et al., 2017;
Krasowsky et al., 2018; Saha et al., 2018) and the results are shown in Fig. 3. The
number fraction values were divided into three different kinds of groups, namely the
roadside, urban or suburban, and background. Results from Fig. 3 show that the
number fractions at the roadside tend to be the lowest. These sites were close to the
traffic sources and the measured BC-containing aerosols were mainly from the traffic.
The left part of the green circles correspond to the relatively clean urban or suburban
sites with the number fractions of the aged BC around 30%. However, the number
fractions of the relative polluted urban or suburban sites had a larger number fraction
of the aged BC around 50%. The number fractions of the aged BC at the background
sites were the largest. Therefore, the number fractions of the aged BC-containing
aerosols were synthetically influenced by the distance from the primary source and the
pollution levels of the ambient atmosphere. The number fraction of the aged
BC-containing aerosols increased with the distance from the primary emission sources
and the pollution levels. Our results were consistent with the aerial measurement by
Metcalf et al. (2012), who found that the number fraction of the aged BC was
29%~41% at the top of the Los Angeles city and 47%-54% for the out plume of this
city.

259       For a better understanding of the source of the fresh and aged BC, we compared

the number concentrations of the BC-containing aerosols with the source
apportionment results from the AMS data. Among the PMF results, the factor of
hydrocarbon-like organic aerosol (HOA) is mainly composed of long-chain
hydrocarbon, and oxygenated organic aerosol (OOA) is mainly from the secondary
formation. HOA is mainly from the diesel exhaust, gasoline exhaust, and lubricating



oil emission. From Fig. 4(a), the number concentration of the fresh BC and mass
concentration of HOA showed good consistency, with $R^2$ equaling 0.69 as shown in
Fig. S6, which further proved the evidence that the fresh BC-containing aerosols were
from the traffic sources. The time series of the aged BC and OOA showed good
consistency as shown in Fig. 4 (b), with $R^2$ equaling 0.87. Therefore, the aging
processing of the ambient BC was accompanied by the ambient OA. The mass
concentration of OOA and number concentration of aged BC can be used as good
indicators for each other.

**3.3 Size distributions of the fresh and aged BC-containing aerosols**

The size distributions of the BC-containing aerosols exert significant influence on
their corresponding radiative effects (Matsui et al., 2018; Zhao et al., 2019). We
calculated the number size distribution (NSD) of BC-containing aerosols for the fresh
and aged ones at different sites, and the results are shown in Fig. 5. It should be noted
that the Dp in Fig. 5 corresponds to the mobility diameter from the DMA. The
BC-containing aerosol NSD was further fit using the log-normal distribution.
As for the fresh BC-containing aerosols, the geometric mean diameters (Dm) were
193, 161, and 162 nm for the PKU, CP, and LJ sites, respectively. The geometric
standard deviations (GSD) of the BC-containing aerosol NSD were 1.50, 1.63, 1.91
for the PKU, CP, and LJ sites, respectively. The GSD to some extent reflects the
diversity of the BC sources. The LJ site had the largest GSD, which indicated multiple
sources of fresh BC-containing aerosols. The LJ site was highly influenced by
atmospheric transportation, due to the high altitude of this location (Zheng et al., 2017;
Tan et al., 2021). Therefore, the fresh BC-containing aerosols could be originated from
different orientations. As for the urban site PKU, the fresh BC aerosols were mainly
from urban lifestyle emissions. Therefore, the fresh BC aerosols at the PKU site had





the lowest value of the GSD. However, the fresh BC aerosols at the suburban site CP
were influenced synthetically by urban lifestyle sources and some other sources from
suburban, and thus had a larger value of GSD than that of PKU.

293         As for the aged BC, it is obvious that they had larger diameters than those of the

fresh BC due to the coating of other non-BC components. The Dm values of the aged
BC were 294, 244, and 257 nm for the PKU, CP, and LJ sites, respectively. The
corresponding GSD values were 1.37, 1.41, and 1.46.

297         Based on the above results, the Dm values of the aged BC aerosols were larger

than that of the fresh BC aerosols by 52%, 52%, and 59% for the PKU, CP, and LJ
sites, respectively. The GSD values of the aged BC were consistent with that of the
fresh BC with the lowest value at the PKU site and highest value at the LJ site, which
is consistent with the diversity of the sources of BC-containing aerosols. For each site,
the GSD values of the aged BC aerosols were smaller than that of the fresh ones. The
GSD of BC-containing aerosols tend to be smaller during the aging processing
because the increment of the diameter should decrease with the diameter.
**3.4 Size distribution of the fresh and aged BC core**

306         The number and mass concentrations of the BC core under different mass

equivalent diameters were calculated with the assumption that the BC core has an
effective density of 1.8 g/cm$^3$ and the results are shown in Fig. 6 and Table 1. From
Fig. 6, the geometric mean mass equivalent diameter ($D_{me}$) of the fresh BC particles
were 115, 107, and 127 nm, for the PKU, CP, and LJ sites respectively. The
corresponding GSD values are 1.58 1.53 and 1.68, respectively. The $D_{me}$ for the aged
BC particles were 114, 95, and 111 nm for the PKU, CP, and LJ sites respectively and
the corresponding GSD values were 1.40, 1.45, and 1.43, respectively. Both the GSD





and the $D_{me}$ of the aged BC were smaller than that of the fresh BC. This might be
resulted from the fact that the small BC particles have a longer life than that of the
large BC particles.

**3.5 Morphology of the BC-containing aerosols**

The time series of the number fractions of the attached BC-containing aerosols to
the total BC- containing aerosols ($f_{attached}$) are shown in Fig. 7. From Fig. 7, the $f_{attached}$
ranged between 0 and 0.21 with a mean value of $7.2 \pm 3.7\%$, $11.0 \pm 3.7\%$, and $10.1 \pm$
$4.1\%$. Moteki et al. (2014) found that the $f_{attached}$ was generally less than 0.1 in Tokyo.
The $f_{attached}$ ranged between 3% and 16% in suburban London (Liu et al., 2015). A
mean value of 12% was found for biomass burning particles using electron
microscopy (China et al., 2013). Our measurement results were consistent with the
previous studies. The $f_{attached}$ tend to increase with the $PM_{2.5}$ for different sites, which
may indicate that the attached BC-containing aerosols were generated from the
coagulation of BC and non-BC aerosols.
We calculated the $f_{attached}$ under different aerosol diameters and the results are
shown in Fig. 8. There were few attached BC-containing aerosols when the diameter
was smaller than 250 nm with $f_{attached}$ lowing than 2%. The $f_{attached}$ increased with the
diameter for all of the measurement sites. It could reach 30% for the LJ sites. Based on
the results from the electron microscopy, the BC volume fractions are smaller than
those of the non-BC volume fractions in the attached BC aerosols (Moteki et al., 2014).
Our results further indicate that the attached BC aerosols were formed from
coagulation, as the coagulation efficiency of the two particles increased with the
difference between their sizes (Kim et al., 2016; Cai et al., 2017; Mahfouz et al.,

337 2021).





The f$_{attached}$ under different aerosol number concentrations (N) and different ratios
of the BC-free aerosol number concentrations to the BC-containing aerosol number
concentrations are shown in Fig. 9. Results showed that the f$_{attached}$ increased with the
above two factors. The results were consistent with the fact that the coagulation
between BC and non-BC components is more likely to happen with the increment of
the BC-free aerosol number concentrations. Based on the analysis above, we
concluded that the attached BC- containing aerosols are mainly formed through
coagulation.
**4 Conclusions**
In this study, the BC microphysical properties were studied based on field
measurement using the DMA-SP2 system at the urban site PKU, suburban site CP and
a background site LJ.
The BC-containing aerosols were sorted as aged thickly coated BC and fresh thinly
coated BC based on the lag time between the peak position of the light scattering
signals and the incandescence signals. The number fractions of the aged
BC-containing aerosols were 49%, 33%, and 79% for the PKU, CP, and LJ sites
respectively. The mass concentrations of the fresh BC-containing aerosols showed
good consistency with that of HOA, which indicated that the fresh BC-containing
aerosols were mainly generated from the emission of vehicles. The aged
BC-containing aerosols are highly correlated with the OOA.
The geometric diameter of the fresh BC-containing aerosols ranged between 160
nm and 200 nm, while the corresponding range was 240~300 nm for the aged
BC-containing aerosols. The GSD of the BC-containing aerosols decreased during the
aging process. The corresponding mobility diameters of these aged (fresh)
BC-containing aerosols were 294 (193), 244 (161), and 257 (162) nm. The measured





aged (fresh) BC core number median diameters were 115 (114), 107 (95), and 127
(111) nm for the urban, suburban, and background sites, respectively. The
corresponding aged (fresh) core mass median diameters were 187 (154), 182 (146),
and 238 (163) nm respectively. The mean diameter of the aged BC-containing aerosols
was larger than that of the fresh BC-containing aerosols, while the mean BC core
diameter of the aged BC-containing aerosols was smaller than that of the fresh
BC-containing aerosols.
The BC-containing aerosols were sorted as the coated type when the scattering
cross-section decreased to zeros, while the BC-containing aerosols were sorted as the
attached type when the scattering cross-section was still larger than a critical point
after passing through the SP2 laser beam. There are about 10% of the BC-containing
aerosols with the BC core attached to the other non-BC components. We concluded
that the attached BC-containing aerosols were mainly generated by coagulation
between the BC and non-BC components.
***Data availability.*** The data is available at https://doi.org/10.5281/zenodo.5816310.
***Author contributions.*** **Gang Zhao:** Conceptualization, Writing - Original Draft,
Visualization, Software, **Tianyi Tan:** Data Curation, Conceptualization, Visualization,
**Shuya Hu**: Data Curation, Conceptualization, **Zhuofei Du:** Data Curation, **Dongjie**
**Shang:** Data Curation, **Zhijun Wu**: Data Curation, Conceptualization, **Song Guo**:
Data Curation, Conceptualization**, Jing Zheng**: Data Curation, Conceptualization**,**
**Wenfei Zhu**: Data Curation, Conceptualization**, Mengren Li**: Data Curation,
Conceptualization**, Limin Zeng**: Data Curation, Conceptualization**, Min Hu**:
Resources, Supervision, Data Curation, Conceptualization, Revision.
***Competing interests.*** The authors declare that they have no conflict of interest.



***Acknowledgments.*** This work is supported by the China Postdoctoral Science
Foundation (2021M700192) and National Natural Science Foundation of China

389 (91844301).

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



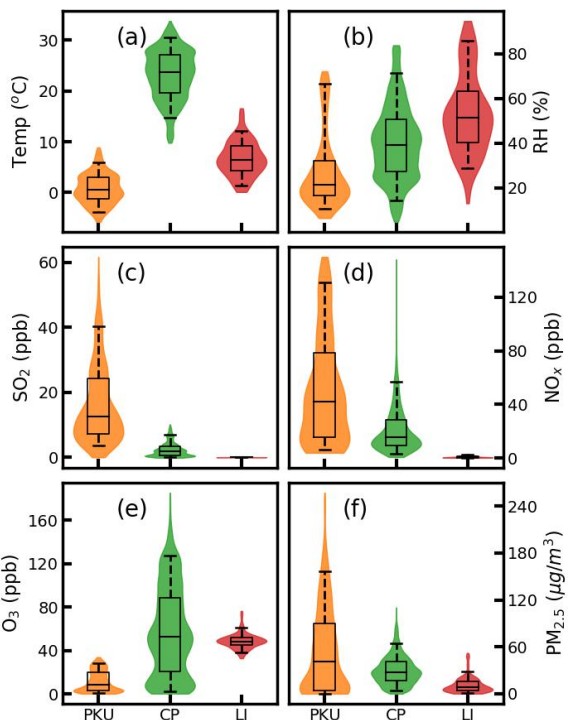


**Figure 1.** The measured distribution of (a) temperature, (b) RH, (c) $SO_2$, (d) NOx, (e) $O_3$ and (f) $PM_{2.5}$ for PKU (orange), CP (green) and LJ (red) sites, respectively. The box and whisker plots represent the 5th, 25th, 75th, and 95th percentiles. The width of the filled colors represents the probability distributions of the corresponding measured values.

622





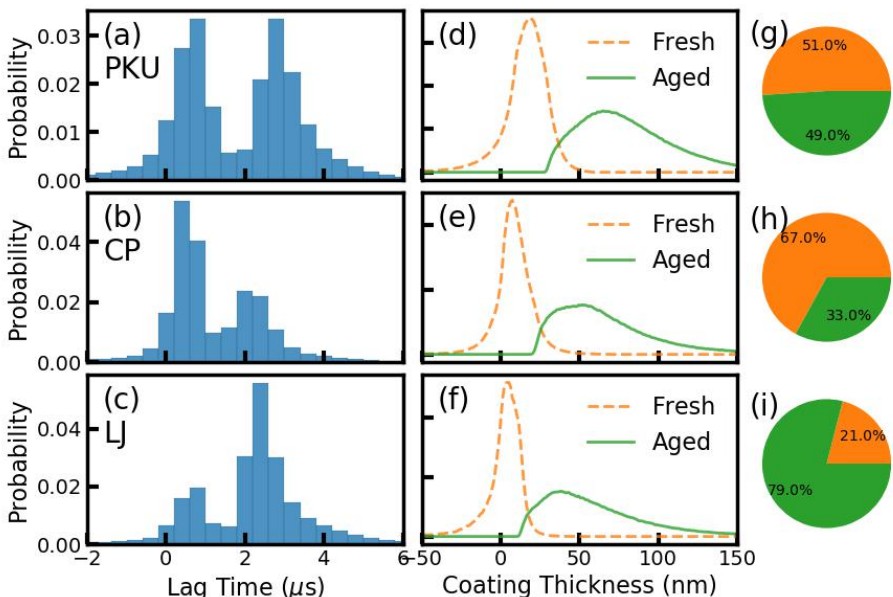

623

**Figure 2.** (a) The measured probability distribution of the lag time for the PKU site. Panel (d) shows the corresponding coating thickness distributions of fresh (orange) and aged (green) BC-containing aerosols. Panel (g) gives the number fraction of the fresh (orange) and aged (green) BC-containing aerosols. Panel (b), (e), and (h) are the corresponding values for the CP site. Panel (c), (f), and (g) give the results for LJ sites.




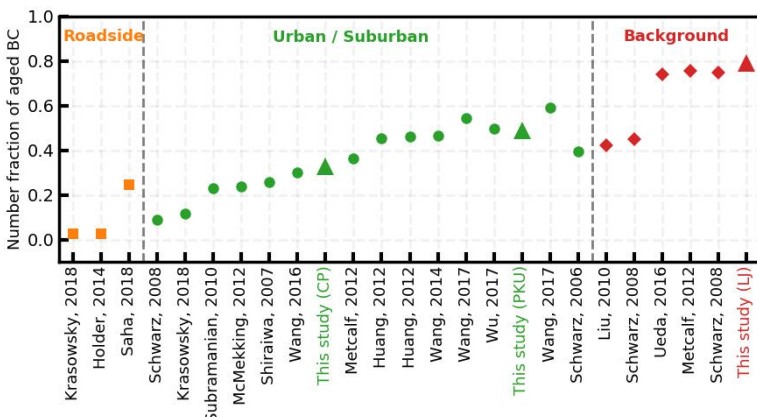


**Figure 3.** Measured number fraction of the aged BC under different atmospheric environments based on literature. Our measured values are shown as triangles.





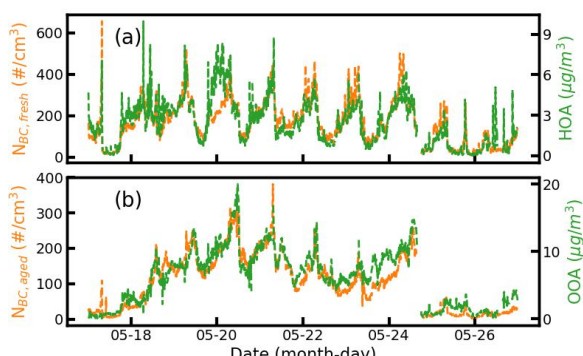


**Figure 4.** The time series of (a) the number concentration of the fresh BC (orange) and the mass concentration of HOA (green), (b) the number concentration of aged BC (orange), and the mass concentration of OOA (green).






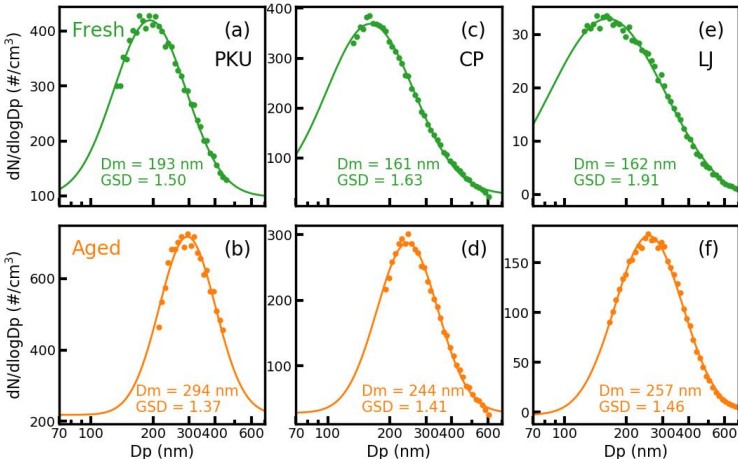


**Figure 5.** The number size distributions of the fresh BC-containing aerosols at (a) PKU, (c) CP, and (e) LJ sites. Panels (b), (d), and (f) are the number size distributions of the aged BC-containing aerosols for the PKU, CP, and LJ sites, respectively. The dots in the figure are the measurement results and the lines are the corresponding fit results with a log-normal distribution.

645



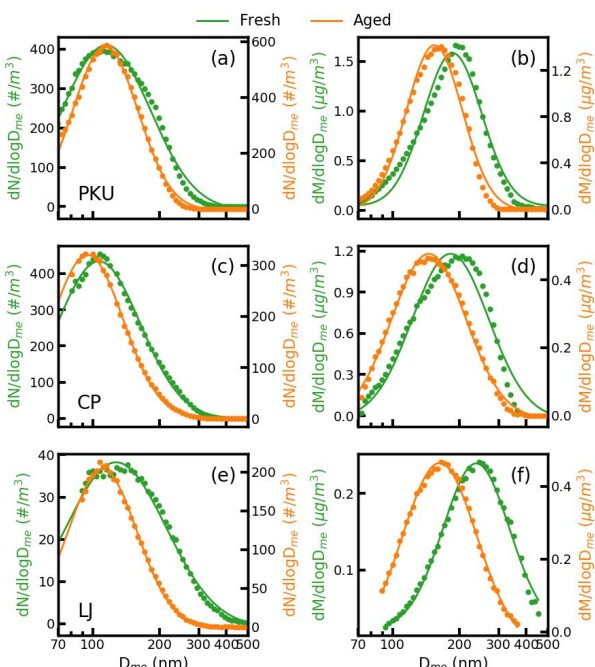

646

**Figure 6.** The BC core number size distributions of the fresh (green) and aged (orange) BC aerosols for the (a) PKU, (c) CP, and (e) LJ sites. Panel (b), (d) (f) show the BC core mass distributions of the fresh (green) and aged (orange) BC aerosols for the PKU, CP, and LJ sites, respectively.

651





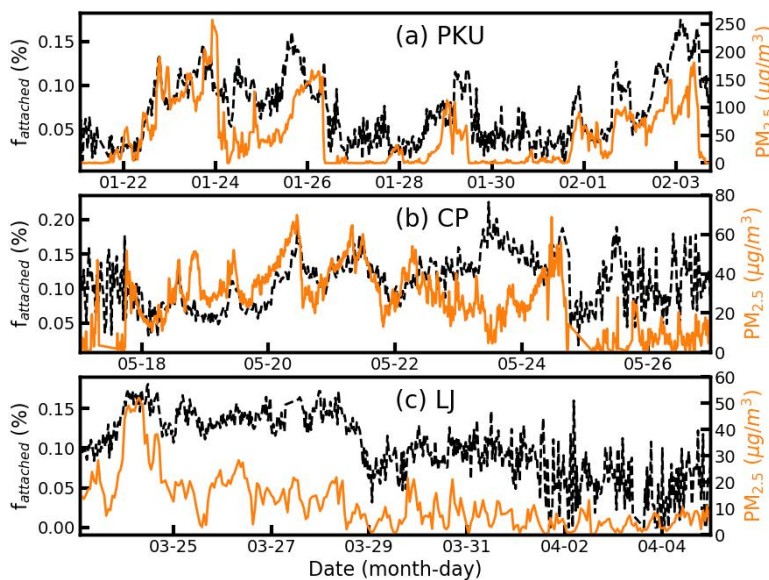

**Figure 7.** The time series of the number fractions of the attached BC (black) and PM$_{2.5}$ mass concentrations (orange) for the (a) PKU, (b) CP, and (c) LJ sites.






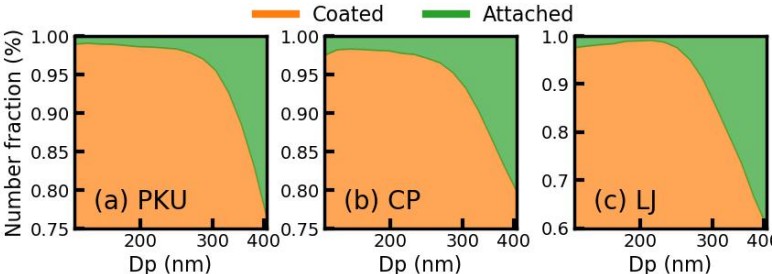


**Figure 8.** The number fractions of the coated and attached BC under different

diameters for the (a) PKU, (b) CP, and (C) LJ sites.






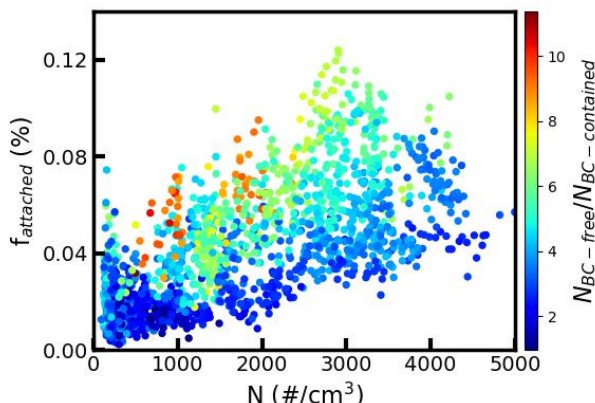


**Figure 9.** The number fractions of the attached BC aerosols under different total aerosol number concentrations for the CP sites. The filled colors represent the ratios between the BC-fee aerosol number concentrations to the BC-containing aerosol number concentrations.

666





667 **Table 1.** The $D_{me}$ and GSD values of the BC core at different sites.

| Site | Value | Number Distribution | | Mass Distribution | |
|---|---|---|---|---|---|
| | | Fresh | Aged | Fresh | Aged |
| CP | $D_{me}$ (nm) | 115 | 114 | 187 | 154 |
| | GSD | 1.58 | 1.40 | 1.35 | 1.34 |
| PKU | $D_{me}$ (nm) | 107 | 95 | 182 | 146 |
| | GSD | 1.53 | 1.45 | 1.48 | 1.47 |
| LJ | $D_{me}$ (nm) | 127 | 111 | 238 | 163 |
| | GSD | 1.68 | 1.43 | 1.47 | 1.41 |

668