# Peer review of "Mixing state of black carbon at different atmospheres in north and southwest"

_Atmospheric Chemistry and Physics, 2022_

## Author Comment (AC1)

Response to reviewer#1

Thanks for the reviewer's helpful professional comments on the manuscript and we agree with the reviewer's helpful suggestions. The point-by-point responses are listed below.

*Comment:* This study presents the measurements of BC microphysical properties at three different sites spanning urban, semi-urban, and rural environments over the North China Plain region. These are potentially valuable datasets to understand the evolution of BC over this anthropogenically influenced region.

However, a few key issues regarding the atmospheric processes of BC have not been analyzed and demonstrated sufficiently. Discussions in many places are loose and not referenced. The technical part, data analysis, and discussion need expansion. I, suggest major revision and list a few suggestions.
*Reply:* Thanks for the comments.

Major points:
*Comment:* 1. The time-lag method of SP2 is associated with large uncertainties, which can only roughly separate the thinly and thickly coated BC, based on the time for the coating to be sufficiently evaporated and display a deceased scattering signal before incandescence. Given you have used the DMA-SP2 setup, I would suggest comparing this metric with the DMA-SP2 derived coating status at different core size ranges.

In addition, the more advanced technique using the coated over core size diameter derived from the SP2 should be acknowledged, particularly over this region on the ground (10.5194/acp-19-6749-2019) and vertical profiles (10.1021/acs.est.9b03722). These coating statuses of BC, rather than only separating the thinly and thickly coated BC, should be compared with your DMA-SP2 results.

At the moment, the DMA-SP2 method and time-lag have been mixed and it is not clear how the coatings have been calculated, and what metric have you actually used. Please check through the texts and make the statement clearer.
*Reply:* Thanks for the comments. We agree with the reviewer that the time-lag method of SP2 is associated with some uncertainties, and thus we changed the classification of BC particles as thinly and thickly coated BC instead of fresh and aged BC in the manuscript.

All of the BC particles were firstly classified as thinly or thickly coated BC particles. Then the coating thickness is calculated. In previous SP2 studies, the whole BC diameter (Dp) usually refers to the optically equivalent diameter of the whole particle diameter, which is derived from the Mie calculation with several presumed input parameters (Taylor et al., 2015). Meanwhile, the BC core diameter (Dc) refers to the volume-equivalent diameter (or mass-equivalent diameter) of the BC core by assuming a fixed density for a void-free spherical BC core (1.8 g/cm$^3$ is widely used). However, the BC cores in ambient BC particles always contain some inside voids and

tend to contain more voids in thinly coated BC particles. Thus, using a density of 1.8 g/cm³ will underestimate the core size and overestimate the coating thickness (and shell = core ratio) to some extent. In this study, with the benefit of the DMA–SP2 coupled system, the Dp can be directly measured by the DMA, which is the mobility diameter of the whole article. To derive the Dc, instead of using the fixed density, we applied the closure study suggested by Zhang et al. (2018) to derive the size-dependently effective density for thinly coated BC particles (Fig. R1 for the LJ site) and adopted a density of 1.2 g/cm³ for thickly coated BC particles (Tan et al., 2021). The multiply charged particles induced by the DMA can be removed by comparing the mobility diameter with the optical diameter derived from the SP2 data. Figure R2 shows the comparison between the mobility diameter and optical diameter of thinly and thickly coated BC for single-charged particles. From Figure R2, the mobility diameter and optical diameter agree well and thus the method above can be used to derive the coating thickness of the thinly and coated BC particles. We also compared the coating thickness distribution of the BC particles of thinly and thickly coated BC particles as shown in Fig. 2 (d), (e), and (f), which further demonstrated the method is adaptive. It should be noted here that, when it comes to the BC size distribution, the mass-equivalent diameter of BC cores (assuming a density of 1.8 g/cm³) was adopted in this study for direct comparison with previous studies.

We have to acknowledge the results from Ding et al. (2019) and Liu et al. (2019) are very adaptive for the ground and vertical profiles of BC particles in Beijing. Some discussions were added to the manuscript.

[Figure]

Figure R1. Size-dependently effective density of the BC core for thinly-coated BC particles.

[Figure]

Figure R2. Comparison between optical diameter and mobility diameter for (a) thinly-coated BC and (b) thickly-coated BC.

*Comment:* 2. The definition of "fresh and aged" BC using the time-lag method may be sometimes questionable, because not like traffic sources, some other sources such as biomass burning (10.1029/2008GL033968) or solid fuel burning (10.5194/acp-14-10061-2014; 10.5194/acp-19-6749-2019) initially have a higher coating but may not be that aged. These should be referenced and discussed. I think the thinly and thickly coated BC is a fair classification for this purpose. Because you have three sites, which are three levels of aging, then you can discuss the fraction of thickly-coated BC at different aging scales at the three sites.

*Reply:* Thanks for the comments. We have changed the classification of BC particles as thinly and thickly coated and some discussions were added in section 2.3 in the manuscript correspondingly.

*Comment:* 3. The method using the magnitude of scattering signal remaining after the particle experiences evaporation in the SP2 laser beam, to indicate the attached or coated type of BC should be detailed. What is the threshold used, and what is the uncertainty. It may be useful to show readers how this has happened, especially for the audience without previous knowledge of the SP2 instrument.

How this is related to the thickly-coated BC is determined by the time-lag method. Are we supposed to see a less fraction of attached BC when more thickly-coated?

*Reply:* Thanks for the comments. We added some descriptions of the method using the magnitude of the scattering signal to distinguish the coated and attached BC particles. At the same time, it should be noted that the BC particles are firstly distinguished as thinly and thickly coated BC, and then the thickly coated BC are further grouped as BC and coated BC.

Figure. R3 gives examples of the measured scattering and incandensce signals for attached and coated particles. For the coated type, all of the coating material will evaporate and the scattering cross-sections will decrease to zero after passing through the laser beam, while the scattering cross-section of the attached BC-containing aerosol will not decrease to zero. Firstly, the beam profile, which indicates the theoretically calculated light scattering time series with the assumption that the coating material would not evaporate when passing through the laser beam. Then the corresponding scattering cross-section time series were calculated ($C_s$ in fig. R3) by comparing the measured scattering signal time series (Scat. In fig.R3) and the calculated beam profile using the method of Dahlköter et al. (2014). These particles are sorted as the attached BC ones when the mean $C_s$ values, of which the time range corresponds to the 10% tail of the beam profile, were larger than 5% of the maximum value of $C_s$.

[Figure]

**Figure R3.** The scattering and incandesce signals for (a) attached and (b) coated particles. The Cs represent the calculated time series of the scattering cross-section. The Beam profile denotes the theoretically calculated scattering signals if the non-BC components were not evaporated.

Some descriptions were added to the manuscript.

*Comment:* 4. Regarding the BC size, firstly, the all-BC core size distribution should be given, and the geometric mean and deviation should be compared with such measurements in this region (10.5194/acp-19-6749-2019). This may be source-related or related to aging time. I would suggest performing the same lognormal fitting on Fig. 6 like in Fig. 5 to obtain the distribution of both core and coated sizes. The core size and coated size should be discussed together and possibly derive a general coating thickness for each site. The BC core distribution associated with each coated size will

be worth showing, in conjunction with and compared with other studies in this region, such as using uncoated and coated sizing to derive optical properties (10.1016/j.chemosphere.2020) and CCN properties (10.1021/acs.est.9b03722).

*Reply:* Thanks for the professional comment. We added the descriptions of all-BC core size distribution in section 3.4 and figure 6 in the manuscript. Some comparisons with the previous measurement were added correspondingly. We performed the same lognormal fitting on Fig. 6 as in Fig. 5. We also agree with the reviewer that the BC core distribution associated with each coated size is worth showing. In this study, we mainly present the measured BC microphysical properties with the classification of thinly coated and thickly coated BC particles. We are inspired by the reviewer's opinion and prepare another manuscript to discuss the BC core distribution for different core and coated sizes, which were source-related or related to the aging time, and to analyze their optical properties and CCN properties.

*Comment:* The possible issue is the defined "aged" BC had a smaller core than "fresh" BC because the time-lag method is quite sensitive to the BC core size, as a smaller BC core will take a longer time to evaporate the coatings on it. This does not mean the BC is definitely "aged" (as aforementioned), but the higher time-lag one just has a smaller core and larger coating/BC core ratio, i.e. a larger relative coating thickness. Indeed, the aged BC should have a larger core if no precipitation is experienced, because of coagulation among BC cores. This seeming contradiction needs explanation.

*Reply:* Thanks for the comments. We changed the classification of the BC particles into thinly coated BC and thickly coated BC following the reviewer's recommendation. We agree with the reviewer that the aged BC should have a larger core if no precipitation is experienced while the coagulation among BC cores exists. However, more than half of the particles in our three measurement sites were those of BC-free aerosols. We think the coagulation was dominated by the BC-containing particles and BC-free particles. There are mainly two reasons that may lead to the smaller geometric mean diameter for the thinly coated BC than the thickly coated BC. First, we agree with the reviewer that the smaller BC core tends to have a higher time lag, and thus the thinly coated particles tend to have smaller core diameters. Second, it takes less time for the smaller BC particles to grow the same amount of coating thickness when the increment of the BC particles was dominated by condensation. Some discussions were added in section 3.4 correspondingly.

*Comment:* About the coated BC size distribution, the rural site even had the smallest size for both modes of BC, compared to the urban site. Shouldn't BC have a larger size after aging from urban to rural sites? This needs explanation.

*Reply:* Thanks for the comments. The differences in the BC core size distribution were mainly related to different sources. We agree with the reviewer that the BC may have a larger size after aging from urban to rural sites considering the coagulation between the BC particles. It should be noted here that, when it comes to the BC size distribution, the mass-equivalent diameter of BC cores ($D_{me}$) (assuming a density of

1.8 g/cm$^3$) was adopted in this study for direct comparison with previous studies. However, if the coagulation between the BC and non-BC particles dominates the coagulation process, the mass equivalent to BC core diameter should keep relatively constant after aging.

*Comment:* It is not careful that sometimes you used Dm or Dp for the coated BC diameter, which should be consistent throughout the texts.
*Reply:* Thanks for the comment. We have changed all of the Dm into Dp in the manuscript。

*Comment:* Section 3.4 needs expansion and is referenced. There have been wide observations that heavy pollution has caused a substantial increase in BC coatings and sizes (10.5194/acp-18-9879-2018; 10.5194/acp-19-6749-2019), which should be compared with your study.
*Reply:* Thanks for the comment. We have expanded the discussion of section 3.4 in the manuscript.

*Comment:* 5. The DMA-SP2 has been used to derive the particle shape of BC-containing particles (10.1029/2021GL094522), which should be referenced to aid your results. The interesting part is you found the attached faction increased with larger Dp, which is essentially consistent with the results in the above reference that larger Dm will contain more fractal BC hard to be enveloped by coatings.
*Reply:* Thanks for the comments and prospective view about our measurement results. We have added the discussion in section 3.5.

*Comment:* 6. The coagulation between non-BC and BC-containing particles is an interesting one. It shows a higher concentration or more polluted condition caused a higher fraction of the attached type. That means heavier pollution will lead to a higher fraction of attached type (not core-shell) BC, this is somewhat contradictory to the normal knowledge that for a more polluted condition, where more secondary aerosol forms, more condensation process could lead to a higher fraction of more coated and spheric BC, so less attached. This needs explanation. How are these related to the aged/fresh BC.
*Reply:* Thanks for the comments. Under the heavier pollution, more secondary aerosol forms and more condensation process would on one hand increase the coating of the previously coated BC particles, which would not increase the number fraction of coated BC, on the other hand, would coating on the attached BC particle and to some content would lead to some transform from the attached BC to coated BC particles. Based on our measurement results, the above process of transformation from attached BC to coated BC may lag behind the process of coagulation between thinly coated BC and non-BC particles, which would lead to the increment in the fraction of attached BC with the pollution levels. Some discussions were added to the manuscript.

*Comment:* 7. The mereological analysis is largely missing, at least air mass back trajectories, RH, etc. should be analyzed to show how these are related to your results.
*Reply:* Thanks for the comments. We have added the back trajectories studies in the supplement and manuscript section 3.1.

Others:
*Comment:* Fig. 3. a study at a remote site also using the time-lag method could be also compared (10.5194/acp-10-7389-2010).
*Reply:* Thanks for the comments. We added a comparison of the time lag with the related reference.

*Comment:* The conclusion needs to be polished to show the key messages.
There are many places where techniques and discussions are mixed, I only list a few places, such as line 308; line 370-376, please also check through the texts to write as methods and discussions are separated.
*Reply:* Thanks for the comments. We have rephrased some of the texts in the manuscript.

*Comment:* The title should use black carbon, not refectory BC, as the mixing state is associated with refractory BC and coatings. "in" different atmosphere. China is too broad, you may need to specify the region.
*Reply:* Thanks for the comments. We have rephrased the title correspondingly.

*Comment:* Line 23-28, there are too many numbers here, which need a tidy-up.
*Reply:* Thanks for the comments. We have removed some descriptions in the corresponding manuscript.

*Comment:* The conclusion about the mixing state of BC is not clear. How to know the mixing is necessarily by coagulation. I probably understand what you mean here, but the attached one only takes a small proportion but most are still driven by condensation.
*Reply:* Thanks for the comments. We added some description in the conclusion part to clarify that these attached BC-containing aerosols were mainly generated by coagulation between the BC and non-BC components even though the aging of the ambient BC aerosols was driven by condensation.

*Comment:* Line 53-54, should point out that the mixing state of BC importantly determines its absorbing properties but Cappa et al. did not find that, and the study found it displayed as two regimes (10.1038/ngeo2901). The statements of the references need correction.
*Reply:* Thanks for the comments. We have revised the reference.

*Comment:* Line 159, this sentence can't be one single paragraph.
*Reply:* Thanks for the comments. We have merged the corresponding sentence of line

159 and the next paragraph into one paragraph.

*Comment:* The unit of Fig. 9 is %?
*Reply:* Thanks for the comment. We have changed the unit in Figures 8 and 9.

*Comment:* Can we make similar plots as in Fig. 9 for the other two sites.
*Reply:* Thanks for the comments. The other two sites have almost the same trends as figure 9 and we hold that giving only one site in Fig.9 is enough for the results.

Ding, S., Liu, D., Zhao, D., Hu, K., Tian, P., Zhou, W., Huang, M., Yang, Y., Wang, F., Sheng, J., Liu, Q., Kong, S., Cui, P., Huang, Y., He, H., Coe, H., and Ding, D.: Size-Related Physical Properties of Black Carbon in the Lower Atmosphere over Beijing and Europe, Environ Sci Technol, 53, 11112-11121, 10.1021/acs.est.9b03722, 2019.

Liu, D., Joshi, R., Wang, J., Yu, C., Allan, J. D., Coe, H., Flynn, M. J., Xie, C., Lee, J., Squires, F., Kotthaus, S., Grimmond, S., Ge, X., Sun, Y., and Fu, P.: Contrasting physical properties of black carbon in urban Beijing between winter and summer, Atmospheric Chemistry and Physics, 19, 6749-6769, 10.5194/acp-19-6749-2019, 2019.

Tan, T., Hu, M., Du, Z., Zhao, G., Shang, D., Zheng, J., Qin, Y., Li, M., Wu, Y., Zeng, L., Guo, S., and Wu, Z.: Measurement report: Strong light absorption induced by aged biomass burning black carbon over the southeastern Tibetan Plateau in pre-monsoon season, Atmospheric Chemistry and Physics, 21, 8499-8510, 10.5194/acp-21-8499-2021, 2021.

Taylor, J. W., Allan, J. D., Liu, D., Flynn, M., Weber, R., Zhang, X., Lefer, B. L., Grossberg, N., Flynn, J., and Coe, H.: Assessment of the sensitivity of core / shell parameters derived using the single-particle soot photometer to density and refractive index, Atmospheric Measurement Techniques, 8, 1701-1718, 10.5194/amt-8-1701-2015, 2015.

Zhang, Y., Su, H., Ma, N., Li, G., Kecorius, S., Wang, Z., Hu, M., Zhu, T., He, K., Wiedensohler, A., Zhang, Q., and Cheng, Y.: Sizing of ambient particles from a Single Particle Soot Photometer measurement to retrieve mixing state of Black Carbon at a Regional site of the North China Plain, Journal of Geophysical Research:

Atmospheres, 123, 12778-12795, doi:10.1029/2018JD028810, 2018.

---

## Author Comment (AC2)

Response to reviewer#2

Thanks for the reviewer's helpful comments on the manuscript. The point-by-point responses are listed below.

*Comment: This manuscript conducted DMA-SP2 measurements at three sampling sites in China and investigated the microphysical properties of BC-containing particles, including mixing state and morphology. The valuable and high-quality measurement data presented in the manuscript will help better understand BC properties in the different atmospheres in China, which is of great interest to the scientific community. Therefore, I recommend this manuscript for publication, as long as the following comments are properly addressed.*
*Reply:* Thanks for the comments.

*Comment: Some sentences in the abstract seem to be redundant. Please try to make it concise.*
*Reply:* Thanks for the comments. We simplified some of the sentences in the abstract.

*Comment: Line 62, I think this classification is scientifically inappropriate. There are also other types of BC particles, such as fresh fractal-shape aggregates, partially coated, etc.*
*Reply:* Thanks for the comments. We have changed the classification of BC particles into thinly and thickly coated ones. For the thickly coated BC, it can be further divided into two morphological types: bare BC on the surface of non-BC particles or partially coated (attached type) and BC embedded within or coated by non-BC components (coated type).

*Comment: In the section of Part 3.1, the authors provide a detailed introduction of the three measurements, including the meteorology, gaseous pollutants, and PM2.5 pollution features. However, such information is not related to the topic of the manuscript, nor any of the discussion in the following parts. It is suggested that this part is simplified and focuses on what is related to the topic.*
*Reply:* Thanks for the comments. We simplified section 3.1 in the manuscript.

*Comment: Line 209, why in this study a different lag time (1.4 us) was used?*
*Reply:* Thanks for the comments. Two log-normal distributions were used for the probability distribution of the lag time for BC-containing particles:

$$\text{PDF}(\Delta t) = \sum_{i=1,2} \frac{A_i}{\sqrt{2\pi}\log(\sigma_{g,i})} exp\left[-\frac{\log(\Delta t)-\log(\Delta t_i)}{2log^2(\sigma_{g,i})}\right],$$

Where $\Delta t$ is the lag time, $A_i$, $\sigma_{g,i}$, $\Delta t_i$ are the scale factor, geometric standard deviation, and geometric mean lag time of mode $i$ respectively. The lag time was determined by calculating the value when the probability distribution values of mode 1 and mode 2 are equal. Figure R1 gives the probability distribution of the lag time

for the LJ site and the critical value of lag time was 1.4 μs. The critical lag time of 1.3 and 1.7 μs were determined for CP and PKU sites, respectively.

Some values were slightly changed accordingly due to the use of different lag times in the manuscript. We replotted figure 2 in the manuscript.

[Figure]

Figure R1. Lognormal fit (two modes) examples of the lag time distribution during the campaign for the LJ measurement site. The green squares, red dashed line, blue dashed line, and orange dashed line represent the measured lag time distribution, the fit results of the lag time distribution, fit results of mode 1, and fit results of mode 2, respectively.

*Comment:* *Line 228, "the CP site is not far from the urban, and thus the fresh BC particles from the traffic contribute a large amount of the total ones." This part confuses me. Why the CP site instead of the urban site (PKU) is dominated by traffic emissions?*

*Reply:* Thanks for the comments. We do not mean that the CP site instead of the urban site (PKU) is dominated by traffic emissions.

The difference in the number fraction of the thickly coated BC particles was synthetically influenced by the ambient pollution levels and the sources of the BC aerosols. The suburban site CP had the largest number fraction of the thinly coated BC particles. The CP site is not far from the urban, and thus the thinly coated BC particles from the traffic contribute a large amount of the total ones. The urban site PKU had a larger number fraction of the thickly coated BC than that of the CP site. This might be resulted from the PKU site being more polluted than the CP site and then the aging processing at the PKU site was faster than that at the CP site.

*Comment:* *Line 332, is electron microscopy data from this study or the previous study (Moteki et al., 2014)? If it is from the previous study, is the conclusion representative of the situation in this study?*

*Reply:* Thanks for the comments. The number fraction of attached BC particles from Moteki et al. (2014) is from the measurement of SP2, which adopted the same method as our studies. Therefore, the number fraction of attached BC particles can be used to compare with our studies.

*Comment:* *Figure 4, which sampling site is discussed here?*

*Reply:* Thanks for the comments. The data in figure 4 corresponds to the CP site. We added descriptions in the corresponding texts.

Moteki, N., Kondo, Y., and Adachi, K.: Identification by single-particle soot photometer of black carbon particles attached to other particles: Laboratory experiments and ground observations in Tokyo, Journal of Geophysical Research: Atmospheres, 119, 1031-1043, https://doi.org/10.1002/2013JD020655, 2014.

---

## Author Response (AR2)

Response to editor:

Thanks for the editor's comments on the manuscript and the point-by-point responses are listed below.

*Comment:* I kindly ask you to answer the question below and also to consider and correct the following indicated typos for the acceptance of the manuscript.
*Reply:* Thanks for the comments.

*Comment:* Question:
How the optical diameter compares so well with mobility diameter when BC is thinly coated in Fig. R2 (author response file), as it shouldn't be spherical.
*Reply:* Thanks for the comments. The color in the figure represents the probability distribution of the optical diameter at a given mobility diameter. As noted in the figure, some uncertainties remain when comparing the mobility diameter and optical diameter of thinly-coated and thickly coated BC particles. If figure R1, the uncertainties of retrieved optical diameter of the thickly-coated BC is smaller than that of the thinly-coated BC particles, which is in agreement with the fact that the thinly coated BC is less spherical.

[Figure]

**Figure R1.** Comparison between optical diameter and mobility diameter for (a) thinly-coated BC and (b) thickly-coated BC.

*Comment:* Typos:
L150-156: Despite, sourced, two, containing.
L167: materials
L355: I believe the word "rather" is missing before "...than the thickly coated BC."
L368: geometric
L369: … comparable to Zhang et al. (2018a)…
L397: would coat
L400: add "be" before comparable.
*Reply:* Thanks for the comment. The typos are corrected correspondingly.

*Comment:*
Rewrite the sentences:
L150-153: That sentence is not readable/understandable.
*Reply:* Thanks for the comment. We have rewritten this paragraph.

*Comment:* L165-167: This sentence is not clear. "Details of distinguish the BC-containing…"? "Refer to section … " which section, suppl or 3.4? Please, clarify.
*Reply:* Thanks for the comment. We have rewritten this paragraph.

***Comment:*** L399: Do you mean transformation or transforming? Please clarify the sentence.

***Reply:*** Thanks for the comment. We mean transformation here and the sentences were corrected correspondingly.